# Goat’s Skim Milk Enriched with *Agrocybe aegerita* (V. Brig.) Vizzini Mushroom Extract: Optimization, Physico-Chemical Characterization, and Evaluation of Techno-Functional, Biological and Antimicrobial Properties

**DOI:** 10.3390/foods14061056

**Published:** 2025-03-19

**Authors:** Danijel D. Milinčić, Ivana Sredović Ignjatović, Dejan Stojković, Jovana Petrović, Aleksandar Ž. Kostić, Jasmina Glamočlija, Ana Doroški Petković, Ana Plećić, Steva Lević, Vladislav Rac, Vladimir B. Pavlović, Slađana P. Stanojević, Viktor A. Nedović, Mirjana B. Pešić

**Affiliations:** 1Institute of Food Technology and Biochemistry, Faculty of Agriculture, University of Belgrade, Nemanjina 6, 11080 Belgrade, Serbia; danijel.milincic@agrif.bg.ac.rs (D.D.M.); isredovic@agrif.bg.ac.rs (I.S.I.); akostic@agrif.bg.ac.rs (A.Ž.K.); ana.doroski@agrif.bg.ac.rs (A.D.P.); ana.bjekovic96@gmail.com (A.P.); slevic@agrif.bg.ac.rs (S.L.); vladarac@agrif.bg.ac.rs (V.R.); vlaver@agrif.bg.ac.rs (V.B.P.); sladjas@agrif.bg.ac.rs (S.P.S.); vnedovic@agrif.bg.ac.rs (V.A.N.); 2Institute for Biological Research, “Siniša Stanković”—National Institute of the Republic of Serbia, University of Belgrade, Bulevar Despota Stefana 142, 11108 Belgrade, Serbia; dejanbio@ibiss.bg.ac.rs (D.S.); jovana0303@ibiss.bg.ac.rs (J.P.); jasna@ibiss.bg.ac.rs (J.G.)

**Keywords:** mushroom, goat’s milk, central composite design, techno-functional properties, antiproliferative properties, antimicrobial activity, anti-inflammatory effect

## Abstract

The aim of this study was to develop a novel functional ingredient—goat’s skim milk enriched with *Agrocybe aegerita* (V. Brig.) Vizzini mushroom extract (ME/M)—using Central Composite Design (CCD). The optimized ME/M ingredient was evaluated for its physico-chemical, techno-functional, biological, and antimicrobial properties. Physico-chemical properties were analyzed using Attenuated Total Reflectance Fourier Transform Infrared (ATR-FTIR) spectroscopy, Scanning Electron Microscopy (SEM), and Dynamic Light Scattering (DLS). The ingredient exhibited a polymodal particle size distribution and contained glucans, along with a newly formed polypeptide resulting from the selective cleavage of goat milk proteins. A 0.1% ME/M solution demonstrated good emulsifying and foaming properties. Additionally, ME/M showed strong antiproliferative effects on human cancer cell lines, particularly Caco-2 (colorectal) and MCF7 (breast) cancer cells. The ingredient also promoted HaCaT cell growth without cytotoxic effects, suggesting its safety and potential wound-healing properties. Furthermore, the addition of ME/M to HaCaT cells inoculated with *Staphylococcus aureus* resulted in reduced IL-6 levels compared to the control (without ME/M), indicating a dose-dependent anti-inflammatory effect. The optimized ME/M ingredient also exhibited antibacterial, antifungal, anticandidal, and antibiofilm activity in one-fourth of MIC. These findings suggest that the formulated ME/M ingredient has strong potential for use in the development of functional foods offering both desirable techno-functional properties and bioactive benefits.

## 1. Introduction

In recent years, the demand for health-promoting and functional food ingredients has continued to grow as modern consumers are increasingly interested in their personal health. For this reason, the current trend in the food industry is focused on producing healthier and sustainable food enriched with natural ingredients or extracts. Edible mushrooms are widely consumed and appreciated worldwide for their medicinal, therapeutic, and nutritional properties, which is also confirmed by the increasing demands and their presence on the market, with a tendency for future growth [1]. Mushrooms are a rich source of carbohydrates (dietary fibers), proteins, minerals, vitamins, and other bioactive compounds [1], making them highly valuable for various applications. The diverse composition allows for widespread use, in particular [2,3]. So far, mushroom powder has been successfully used for the formulation of ready-to-eat and ready-to-cook food products, such as various baked goods (bread, biscuits, or cakes), breakfast cereals, soups, pasta, or noodles [3,4,5,6,7,8,9,10], contributing to some health and technological benefits of the prepared products. In addition, bioactive compounds from mushrooms can be classified into low-molecular-weight (lactones, terpenoids, alkaloids, and phenolics) and high-molecular-weight (homo-, hetero-polysaccharides, glycoproteins, glycopeptides, proteins) bioactive compounds, with a focus on beta-glucans [11,12,13]. These bioactive compounds have a wide range of health-promoting properties such as antioxidant, antimicrobial, anticancerogenic, anti-inflammatory, antidiabetic, and anti-hypercholesterolemic effects, as well as hepatoprotective and immunomodulatory effects [1,11,12,13,14]. Due to the nutritional and functional benefits of their bioactives, mushroom extracts are considered promising dietary supplements and ingredients for the development of healthy food products [12]. However, due to their specific sensory attributes (smell, taste, and color), proteolytic activity, low bioavailability, or water-binding properties [2,15,16], mushroom extracts are insufficiently used in food formulations, resulting in most cases in an uncharacteristic and damaged consistency/structure to the products and an undesirable aftertaste.

On the other hand, milk proteins have a high nutritional value and good sensory and techno-functional properties, which makes them very suitable ingredients for the formulation of various dairy and food products. In recent years, interest in goat milk has increased due to its unique nutritional and functional properties [17,18]. In addition, goat′s milk has many advantages over cow’s milk: it contains valuable bioactive peptides and oligosaccharides, conjugated linoleic acid, has lower allergenicity and better digestibility, and Ca-absorption [17,18,19]. To date, several goat′s milk-based functional products have been formulated and characterized as containing giloy juice [20], medicinal plant extracts [21], *Agaricus blazei* extracts [13], grape pomace seed extracts [22,23], and monofloral pollen [24]. Since goat’s milk proteins lack β-glucan, enriching them with mushroom extracts offers a promising approach for developing innovative functional food products and ingredients. However, the use of glucan-rich mushroom extracts in milk-based products is limited due to insufficient data on their effects on milk matrices [2]. To achieve an optimal product with the best functional, techno-functional, and sensory properties, further optimization is required [25,26,27]. The formulation and optimization of products should include the planning and execution of experiments using a statistical approach called the design of experiments (full/fractional factorial design, response surface methodology, or multiple response optimization). Considering the importance of experimental design in food formulation, the aim of this study was to apply response surface methodology (RSM) and find the optimal mixture of skimmed goat′s milk and the *Agrocybe aegerita* (V. Brig.) Vizzini aqueous extract to obtain functional ingredients with the best antioxidant properties. *A. aegerita* mushroom is widely appreciated for its pleasant taste. In our previous studies, we demonstrated that it possesses a favorable chemical profile (phenolics, tocopherols, and polyunsaturated fatty acids) and good bioactive properties, particularly with respect to their antimicrobial and antibiofilm effects, but we also proved compatibility with dairy products [28,29]. For this study, we used wild-growing mushrooms because, as the extensive literature suggests, they offer a more promising chemical profile and enhanced bioactivities [30,31] compared to cultivated varieties. However, since *A. aegerita* can be commercially cultivated, large-scale incorporation into functional food products is facilitated using cultivated fruiting bodies with standardized chemical characteristics.

Central Composite Design was used to efficiently collect data for the construction of the response surface of five different antioxidant assays: total phenolic content (TPC), antioxidant activity (ABTS^•+^, DPPH^•^ and FRP assays), and chelating ability (CHE), with the pH and mushroom extract content as independent variables. The optimized mushroom–milk ingredient was physico-chemically characterized (glucan content, electrophoretic analysis, ATR-FTIR, and DLS measurements), including the evaluation of its techno-functional (emulsifying and foaming properties, and water/oil holding capacity), biological (antiproliferative, scratch-wound healing, and anti-inflammatory) and antimicrobial properties. This study offers a promising model for formulating optimal ingredients or products that have the best functional properties.

## 2. Materials and Methods

### 2.1. Mushroom Extract and Thermally Treated Goat Milk Samples

Our previous study has shown that the aqueous extract of *Agrocybe aegerita* (V. Brig.) Vizzini has a high content of glucans (total, α- and β-glucans), is a rich source of low-molecular-weight proteins/peptides, and has good functional properties (antioxidant, wound-healing, anti-inflammatory, and antimicrobial characteristics) [13]. For these reasons, the aqueous extract of *A. aegerita* was selected for the preparation of a new functional ingredient based on goat’s milk and a mushroom extract. The preparation of the aqueous extract of *A. aegerita* has already been reported in detail by Petrović et al. [13]. Fresh goat’s milk was collected from a local farm and immediately skimmed by centrifugation. Briefly, raw goat’s milk was incubated in a water bath at 30 °C for 30 min, then centrifuged at 3000× *g* for 30 min at 5 °C, and then placed in a water–ice bath for 30 min. The solidified fat was then removed, skimmed milk was thermally treated (90 °C, 10 min) as previously described by Pešić et al. [32], and it was then spray-dried and further used for the preparation of the optimized ME/M ingredient. Thermally treated goat skim milk (M) without mushroom extract was used as the control sample.

### 2.2. Central Composite Design (CCD)—Methodology

Experimental design is the methodology of conducting and planning experiments in order to extract a maximum amount of information from the data with as few experiments as possible. With the help of experimental design, the most influential factors (variables and parameters), the synergism between the factors, and the optimal conditions can be determined [33]. The two main applications of experimental design are screening (the identification of factors influencing the experiment) and optimization (finding optimal conditions for an experiment) [34]. To test the influence of pH and mushroom extract on the functional properties of the functional ingredient, these two factors were investigated using response surface methodology (RSM). RSM is a valuable tool with which to determine the settings of the experimental factors and evaluate the relationship between the controlled experimental factors and the observed results [35].

The influence of the experimental parameters was investigated using the CCD as an approach for the response surface methodology. The effects and interactions of two independent variables on the total phenolic content (TPC), antioxidant activity (ABTS^•+^, DPPH^•^ and FRP assays), and chelating ability (CHE) of the prepared functional ingredients were investigated to determine the conditions when a maximum of dependent variables could be achieved. The factors investigated were tested at three levels (low, medium, and high). The levels of the factors in the CCD for mixtures of goat’s milk and mushroom extract are shown in Table 1.

The matrix for the analyzed factors was created according to the selected Central Composite Design using MINITAB software (Release 16, Minitab Inc., State College, PA, USA). To ensure that uncontrolled factors did not influence the results, the runs were performed randomly. A full factorial design was conducted with a total of 13 experimental runs, with 4 cube points, 5 central points in the cube, and 4 axial points (Table 2). The experimental measurements of all dependent variables (TPC, ABTS^•+^, DPPH^•^, FRP, and CHE) were performed in triplicate, and the average was taken as the response.

#### 2.2.1. Preparation of ME/M Mixtures

Spray-dried thermally treated goat milk powder (see Section 2.1) was reconstituted with milliQ water (1:10 *w*/*v*, 10%). The reconstituted milk was divided into three glasses to adjust the desired pH (7.0; 6.25; and 5.50). Then, the suggested amounts of lyophilized mushroom extract were added to the pH-adjusted milk to prepare 13 mixtures (Table 2). The mixtures were stirred on a mechanical shaker for 1 h and then sonicated for 30 min to achieve optimal solubilization.

#### 2.2.2. Total Phenolic Content and Antioxidant Properties of Prepared Mixtures

Briefly, to determine the total phenolic content, 35 μL of the appropriately diluted samples was mixed with the Folin–Ciocalteu reagent (150 μL) and 7.5% sodium carbonate (115 μL). The reaction mixture was allowed to react for 1 h 30 min in a dark place, and the absorbance was recorded at 765 nm. TPC was expressed as mg gallic acid equivalents per 100 mL (mg GAE/100 mL). Antioxidant assays, such as the ferric-reducing power (FRP), ABTS^•+^ scavenging activity (ABTS^•+^), DPPH^•^ scavenging activity (DPPH^•^), and ferrous ion-chelating capacity (CHE) of the prepared mixtures were evaluated as previously described [23,24,36]. For the ABTS^•+^ assay, 15 μL of the sample was mixed with 300 mL of the ABTS^•+^ working solution. After 7 min of mixing, the absorbance was measured at 734 nm. For the DPPH^•^ assay, 40 µL of the sample was mixed with 260 µL of the DPPH^•^ working solution, and after incubation in the dark, the absorbance was measured at 515 nm after 30 min. Ferric-reducing power (FRP) was determined by mixing the sample (250 μL) with 250 μL of a 0.2 M phosphate buffer, pH 6.6, and 250 μL of a 1% potassium ferricyanide solution. The mixture was then incubated at 50 °C for 20 min. Then, 250 μL of 10% TCA was added, and the mixture was centrifuged at 17,000× *g* for 5 min. After that, 135 μL of the supernatant was mixed with 135 μL of milliQ water and 30 μL of 0.1% ferric chloride. After 10 min, the absorbance was measured at 700 nm. The results for both radical scavenging assays and FRP were expressed as mg Trolox equivalents per 100 mL of the mixture (mg Trolox/100 mL). A well-known method with ferrozine was used to assess the ferrous ion-chelating capacity (CHE). The results for CHE were expressed as mg EDTA equivalents per 100 mL of the mixture (mg EDTA/100 mL). A microplate reader was used for all the measurements (ELISA Plate Reader).

#### 2.2.3. Preparation of the Optimized ME/M Mixture—Overall Desirability

According to the responses obtained from previous experiments, the overall desirability and optimum composition of the ME/M mixture have been determined. The optimized mushroom content was mixed with 10% milk at an optimized pH. After that, the optimized ME/M mixture was lyophilized and used for further physico-chemical characterization and evaluation of techno-functional, biological, and antimicrobial properties.

### 2.3. Physico-Chemical Characterization of Optimized ME/M Ingredient

#### 2.3.1. Glucan Content

The glucan content (total, α-, and β-glucans) in the optimized ME/M ingredient and M powder was determined using the Megazyme β-Glucan Assay Kit (Yeast and Mushroom), which is suitable for the indirect measurement of 1.3:1.6-β-glucan in yeast and mushroom preparations (Product code: K-YBGL). The procedure was performed according to the manufacturer’s guidelines. The obtained values were calculated using the Mega-Calc™ software tool available on the raw data processing website and expressed as a percentage (https://www.megazyme.com/beta-glucan-assay-kit (accessed on 17 March 2025)).

#### 2.3.2. Electrophoretic Analysis

Sodium dodecyl sulfate-polyacrylamide gel electrophoresis (SDS-PAGE) under reducing conditions (SDS-R-PAGE) was used to characterize the protein profile of the optimized ME/M ingredient and M powder, as previously described in detail [23,32]. SDS-R-PAGE was performed with separating gels (12.5% *w*/*v*; pH = 8.85) and stacking gels (5% *w*/*v*; pH = 6.8), and with a Tris-glycine running buffer [0.05 M Tris (pH = 8.5), 0.19 M glycine, 0.1% *w*/*v* SDS]. Samples were prepared by dissolving 4 mg of lyophilized ME/M ingredient and M powder in an appropriate sample buffer consisting of 0.055 M Tris-HCl (pH = 6.8); 2% (*w*/*v*) sodium dodecyl sulfate (SDS); 7% (*v*/*v*) glycerol; 0.0025% (*w*/*v*) bromophenol blue; and 5% β-mercaptoethanol. Aliquots of 25 µL were loaded into the wells. After analysis, the gels were stained with Coomassie blue dye for 1 h, then destained, scanned, and analyzed using SigmaGel software (SigmaGel software version 1.1, Jandal Scientific, San Rafael, CA, USA).

#### 2.3.3. ATR-FTIR Analysis

The optimized ME/M ingredient and M powder were recorded by an IRAffinity^−1^ spectrometer equipped with an ATR unit (Shimadzu, Kyoto, Japan). The spectra were collected in the wavenumber range of 4000–600 cm^−1^, at a resolution of 4 cm^−1^, from 100 scan accumulations.

#### 2.3.4. DLS Measurements

The particle size of the optimized ME/M ingredient and the control M powder was determined by the Dynamic Light Scattering (DLS) measurement using a Horiba NanoPartica SZ-100 device (Horiba, Japan) [15]. Prior to analysis, both samples were reconstituted in milliQ-water to prepare 0.1% solutions. Measurements were conducted at 25 °C. Size measurements were performed in the polydisperse mode. Each sample was measured in five replicates.

### 2.4. SEM of Optimized ME/M Ingredient

The morphology of the optimized ME/M ingredient and M powder was imaged using a Scanning Electron Microscope (JEOL JSM-6390LV, Tokyo, Japan) at an accelerating voltage of 30 kV, as described by Milinčić et al. [22]. Prior to imaging, the samples were attached to metallic stubs and coated with a gold layer using sputter-coating for 100 s at 30 mA within a BALTEC SCD 005 sputtering chamber (New York, NY, USA).

### 2.5. Techno-Functional Properties of Optimized ME/M Ingredient

Emulsifying properties were determined according to the method described by Wen et al. [37]. In brief, pure sunflower oil (3 mL) and 9 mL of the samples (0.1 g/100 g ME/M and M aqueous solutions, at pH 6.7) were placed in a beaker, and the volume *V*_1_ was recorded. The sample mixture was then homogenized at 10.000 rpm for 1 min, and the volume of emulsion was recorded as *V*_2_. After 10 min, the volume of the emulsion was recorded as *V*_3_, and after 30 min, it was recorded as *V*_4_. The emulsion activity index (EAI) and emulsion stability index (ESI) were calculated as follows:$$EAI=\frac{V_{2}}{V_{1}}$$$$ESI10min=\frac{V_{3}}{V_{1}}$$$$ESI30min=\frac{V_{4}}{V_{1}}$$

Foaming properties were determined according to the method described by Wen et al. [37]. In brief, 15 mL solutions (0.1 g/100 g ME/M and M aqueous solutions at pH 6.7) recorded as A_1_ were mixed at 10.000 rpm for 1 min, and the volume of the sample was recorded as *V*_2_. After 10 min, the volume of the sample was recorded as *V*_3_. Foaming properties were calculated and expressed in a percentage as the foam capacity (FC) and foam stability (FS) according to the following formulas:$$FC=\frac{V_{2}-V_{1}}{V_{1}}*100$$$$FS=(V_{3}-V_{1})*100$$

### 2.6. Biological Properties of Optimized ME/M Ingredient

#### 2.6.1. Antiproliferative Effect on Human Cancer Cell Lines

The crystal violet assay was used to determine the antiproliferative effect. The antiproliferative effect of the optimized ME/M ingredient was analyzed in four different human cancer cell lines: the breast cancer cell line (MCF-7), the cervical carcinoma cell line (SiHa), the hepatocellular carcinoma cell line (HepG2), and the colon adenocarcinoma cell line (Caco-2). The MCF-7, SiHa, and Caco-2 cell lines were grown in high-glucose Dulbecco’s Modified Eagle Medium (DMEM) supplemented with 10% fetal bovine serum (FBS), 2 mM L-glutamine, and 1% penicillin and streptomycin (Invitrogen) at 37 °C in 5% CO_2_. Hepatocellular carcinoma cells HepG2 were cultured in low-glucose DMEM supplemented with 10% FBS, 1% nonessential amino acids and 1% penicillin and streptomycin at 37 °C in 5% CO_2_. Twenty-four hours before the treatment with the ME/M sample, 10^4^ cells/well were seeded in a 96-well plate. After the medium was removed, a fresh medium supplemented with different concentrations of the ME/M (6.25–400 μg/mL) dissolved in phosphate-buffered saline (PBS) was added to the cells. Control cells were grown in a medium that contained the same percentage of PBS that was used for the treatment with the highest concentration of the ME/M sample. The experiment was performed in triplicate for each condition, and cells were incubated with the ME/M sample for 48 h. After that period, the medium was removed, and the cells were washed twice with PBS, stained with a 0.5% crystal violet staining solution, and incubated for 15 min at room temperature. Afterwards, crystal violet was removed, and the cells were washed in a stream of tap water and left to air-dry at room temperature. The absorbance of dye dissolved in methanol was measured in a microplate reader at 570 nm (OD_570_). The results were expressed as IC_50_ (%) of the value in μg/mL. The criterion used to categorize the antiproliferative activity of ME/M to cancer cell lines was as follows: IC_50_ ≤ 20 µg/mL = highly cytotoxic; IC_50_ ranged between 31 and 200 µg/mL = moderately cytotoxic; IC_50_ ranged between 201 and 400 µg/mL = weakly cytotoxic; and IC_50_ > 401 µg/mL = no cytotoxicity.

#### 2.6.2. Cytotoxicity on Non-Tumor HaCaT Cells

The cytotoxicity of the optimized ME/M ingredient towards non-tumor keratinocyte cells (HaCaT) was determined to spontaneously immortalize the keratinocyte cell line, as previously reported in detail [13], using the crystal violet assay. The ME/M ingredient was dissolved in phosphate-buffered saline (PBS) to the working concentration of 8 mg/mL. The results were expressed as the relative growth rate of the cells at different ME/M concentrations (6.25 to 400 µg/mL) compared to the untreated control.

#### 2.6.3. Scratch-Wound Healing Assay

The ability of the ME/M sample to undergo scratch-wound healing was tested. The applied assay, as well as the growth and preparation of HaCaT cells, were described in detail by Petrović et al. [13]. The tested concentration of ME/M was 100 µg/mL. An untreated control without ME/M was also prepared and tested for wound closure (see Section 3.1).

#### 2.6.4. Anti-Inflammatory Properties

The modulation of IL-6 levels in HaCaT cells in response to bacteria and the ME/M sample was measured and used to evaluate the anti-inflammatory properties of optimized ingredients, as previously described by Petrović et al. [13]. Briefly, HaCaT cells were grown on 6-well plates with an adhesive bottom. After removing the medium, fresh fetal bovine serum (FBS) containing 100 µg/mL of the ME/M sample was added to the plate and incubated at 37 °C for 15 min in a 5% CO_2_ incubator. After that, 100 µL of the *S. aureus* culture (10^8^ CFU/mL) was added to the wells, and the mixture was incubated at 37 °C for an additional 4 h. The obtained supernatants were used for the determination of the IL-6 level by the Human IL-6 ELISA Kit (Invitrogen, Vienna, Austria), according to the manufacturer’s procedure. The level of IL-6 was determined in untreated HaCaT cells, HaCaT cells inoculated with *S. aureus* and HaCaT cells treated with the ME/M sample.

### 2.7. Antimicrobial Activity Assays of Optimized ME/M Ingredient

For antimicrobial testing, the following bacteria strains and micromycetes were used: *Staphylococcus aureus* (ATCC 11842), *Bacillus cereus* (food isolate), *Listeria monocytogenes* (NCTC 7974), *L. monocytogenes* (ATCC 13932), *L. monocytogenes* (ATCC 15313), *L. monocytogenes* (ATCC 19111), *L. monocytogenes* (ATCC 35152) (Gram +); *Escherichia coli* (ATCC 25922), *E. coli* (ATCC 11775), *E. coli* O157:H7 (ATCC 700728), *E. coli* O157:H7 (ATCC 43888), *Enterobacter cloacae* (ATCC 45040), *Salmonella Typhimurium* (ATCC 14411), *Y. enterocolitica* (ATCC 9610) (Gram −); *Aspergillus fumigatus* (ATCC 9197), *A. niger* (ATCC 8275), *A. versicolor* (ATCC 11740), *Penicillium funiculosum* (ATCC 48849), *P. verrucosum* var. *cyclopium* (food isolate), Trichoderma viride (IAM 5081) (micromycetes); *Candida albicans* (475/15), *C. albicans* (14/15), *C. albicans* (17/15), *C. parapsilosis* (ATCC 22019), *C. tropicalis* (ATCC 750), and *C. krusei* (H1/18) (micromycetes and candida strains). The tested bacterial and fungal strains were deposited at the Mycological Laboratory, Department of Plant Physiology, Institute for Biological Research, “Sinisa Stanković”, National Institute of the Republic of Serbia, University of Belgrade, Serbia.

The antibacterial, antifungal, and anticandidal assays were described in detail by Petrović et al. [13]. The antibacterial and antifungal activity of ME/M samples were tested using the microdilution method. For antibacterial activity, the ME/M sample was dissolved in 30% ethanol, while for antifungal activity, it was dissolved in 5% DMSO. Before antifungal testing, spores were washed from the surface of the agar plates. Minimum inhibitory/mycelial growth concentrations (MICs), minimum bactericidal concentrations (MBCs), and minimum fungicidal concentrations (MFCs) were determined. The positive controls used were E211 and E224 (for the tested bacteria strains) and ketoconazole (for the Candida strains).

#### Inhibition of Biofilm Formation

The ability of the optimized ME/M sample to inhibit biofilm formation was determined according to the procedure previously described in detail by Petrović et al., [13]. Minimum inhibitory concentrations (MICs), and sub-inhibitory concentrations (½ MIC and ¼ MIC) were determined.

### 2.8. Statistical Analysis

All analyses for the optimized ME/M ingredient (proximate composition, techno-functional, biological, and microbiological properties) were performed in triplicate. Significant differences between means were evaluated by Student’s *t*-test at *p* < 0.05 (StatSoft Co., Tulsa, OK, USA). Graphs were prepared using GraphPad Prism 6 software (San Diego, CA, USA).

## 3. Results and Discussion

### 3.1. Central Composite Design

Response surface methodology (RSM) comprises a group of mathematical relationships that relay the fit of empirical models to the experimental data acquired in a factorial experimental design [38]. The experimental data were represented using a second-order polynomial equation (Equation (1)) which considers the linear, square, and interaction terms, as follows:(1)$$y=\beta_{0}+\sum_{i=1}^{k}\beta_{i}x_{i}+\sum_{i=1}^{k}\beta_{ii}x_{i}^{2}+\sum_{i\leq 1j\leq 1}^{k}\beta_{ij}x_{i}x_{j}+\varepsilon$$
where *y* is the response; *x_i_* and *x_j_* are independent variables (*i* and *j* can range from 1 to *k*); *β*_0_ is the model intercept coefficient; *β_i_* represents the intersection coefficients of linear, quadratic, and interaction effects, respectively; *k* is the number of independent parameters (*k* = 2 in this study), and *ε* is the error [39]. The experimental results were statistically analyzed. The confidence interval of each individual factor and their combinations at the 95% confidence level were evaluated.

The relative magnitudes of the effects of the process variables were determined on the basis of the factor effect and the corresponding *p*-value. The estimated effects and regression coefficients are shown in Appendix A, and the ANOVA test was used to determine how the studied factors affected TPC in the prepared functional products. A comparison of the estimated *p*-values with the established criterion (*p* = 0.05) revealed that there were no linear, quadric, or interaction terms that significantly affected the TPC in the analyzed mixtures. The low coefficient of variation (*R*^2^ = 56.09%) and significant lack of fit (*p* = 0.010) indicated that there was no correlation between the response variable and the independent factors. The response surface plot presented in Figure 1a shows that the highest total phenolic content (TPC) was observed in the mixture prepared at the highest values of the independent variables (pH and *w* (ME)) examined.

The data presented in Appendix A were used to evaluate the effect of pH and the mushroom extract on the antioxidant activity estimated by the ABTS^•+^ (ABTS^•+^ radical scavenging activity) assay of the prepared mixtures. The data presented show that the statistically significant variable was pH. The coefficient of variation was *R*^2^ = 79.19%, indicating a good degree of correlation between the response variable and the independent factors. The significance of the linear regression was confirmed via analysis of variance and *p* < 0.05. The linear increase in antioxidant activity with the increasing pH at which the functional products were prepared was evident from the response surface plot (Figure 1b). The effect of the pH value on ABTS^•+^ radical scavenging activity was more pronounced at higher levels of the mushroom extract.

Appendix A summarizes the estimated regression coefficients and analysis of variance for the DPPH^•^ radical scavenging activity response design. In contrast to TPC and ABTS^•+^, for the DPPH^•^ antioxidant activity, the linear and quadric terms of the mushroom extract were found to be significant for the system’s response. This was determined according to *p*-values. The high correlation coefficient *R*^2^ = 95.61% indicates a high degree of correlation between the response variable and the independent factors, as well as a high degree of fit. The presence of linear and quadratic regression can be clearly observed on the response surface plot shown in Figure 1c. The DPPH^•^ radical scavenging activity increased linearly with the increase in the mushroom extract content (*w* (ME)) and reached its maximum and curvature at *w* (ME) = 1% (*m*/*m*).

The influence of the analyzed parameters on the antioxidant activity tested with the ferric-reducing power assay was estimated by comparing the *p*-values with the established criteria (Appendix A). As in the case of the ABTS^•+^ test, the linear and quadric terms for the mushroom extract content were statistically significant. The ferric-reducing power was highly correlated with the investigated parameters (*R*^2^ = 96.00%). The response surface plot (Figure 1d) showed the effects of pH and the mushroom extract content on the tendency to reduce ferric ions. From this plot, it can be seen that the antioxidant activity increases by increasing the ME content with curvature (slight decrease) for the mean pH value of the functional product studied.

According to the results shown in Appendix A, the ferrous-chelating capacity of the investigated mixtures was statistically influenced by the linear term of the mushroom extract content and the quadric terms of the two independent factors investigated. The high correlation coefficient (*R*^2^ = 95.50%) shows a high degree of fitting between the chelating properties and the variable parameters (pH and *w* (ME)). It is important to emphasize that both quadric terms were not only statistically significant but that both negatively affected the analyzed response parameter. This is also evident from the response surface plot shown in Figure 1e. The maximum chelating properties were observed for products with the highest mushroom extract content and a pH of 6.25.

#### Overall Optimization—Desirability Function

A total of five response variables were monitored using response surface optimization. The regression equations, obtained according to Equation (1) and the data listed in Appendix A, as well as the correlation coefficients, are summarized in Table 3. From these data, it can be concluded that TPC was not correlated with the investigated parameters, and ABTS^•+^ was satisfactorily correlated with the linear term of the pH. The results for DPPH^•^ and FRP assays were highly correlated with both the linear and quadric terms of the mushroom extract content, while results for the chelating ability, in addition, were also correlated with the quadric term of pH. The desirability function was used to determine the optimal conditions for the factors examined to define the optimal quality levels for different responses [40].

The desirability function is a quick transformation of different responses to one objective function [41]. The desirability function has two steps: (1) the transformation of every individual response to an individual desirability function (*d*_i_) that ranges from 0 to 1 (*d*_i_ = 0 undesirable response; *d*_i_ = 1 desirable response) and (2) the calculation of overall desirability (*D*) by taking the geometric average of all individual desirability values.

To obtain overall desirability, all five responses (TPC, ABTS^•+^, DPPH^•^, FRP, and CHE) were considered, and the importance of each variable in relation to the others was selected. An importance value of one was selected for all five responses. Weights were then assigned to the individual responses. The weights determine how important it is for *di* to be close to the maximum or minimum. The weights were chosen to be a value of one. TPC was set as the target value, as the correlation coefficient was low with a significant lack of fit, while for four other responses, the maximum was set as the target goal. The target, lower, and higher values were chosen from the results obtained for the response (Table 4).

According to the data set obtained, the overall desirability was *D* = 0.8741 for the mixture prepared by adding 0.709% (*m*/*m*) of the mushroom extract in 10% milk at pH 6.62. The optimized mixture (0.709% ME in 10% M at pH 6.62) was prepared to verify the predicted value. The predicted individual responses and the values obtained after performing antioxidant tests with the optimized ME/M mixture are shown in Table 5. The recovery was calculated as the ratio between the obtained and the predicted value.

An excellent agreement between the predicted and obtained results was achieved only for the FRP antioxidant activity. The lowest experimental data were found for the antioxidant activity measured with the DPPH^•^ assay. Significantly higher experimental than predicted results were found for the total phenolic content and ABTS^•+^ activity, for which there was either no or an acceptable correlation with the investigated parameters, respectively. Contrasting results for TPC and DPPH can be explained by their different mechanisms of action. In the case of complex food matrices such as the ME/M ingredient, the TPC method measures the total reduction capacity of the samples, and the Folin–Ciocalteu reagent can be reduced by many non-phenolic compounds. On the other hand, the assessment of DPPH radical scavenging activity is based on the ability of the ME/M ingredient to donate hydrogen ions and neutralize free radicals.

### 3.2. Phisyco-Chemical Characterization of the Optimized ME/M Ingredient

#### 3.2.1. Glucan Content

The results for the glucan content in the ME/M ingredient and M powder are shown in Table 6. As expected, the ME/M sample contained glucans from the *A. aegerita* mushroom extract, while no glucans were detected in the control M powder. Our previous study showed a high glucan content in the aqueous extract of *A. aegerita* [13]. The total glucan content was 26.62% (*w*/*w*), while the individual α-glucan and β-glucan contents were 1.94% (*w*/*w*) and 24.67% (*w*/*w*), respectively. These results indicate that optimized ME/M is a rich source of glucans, especially β-glucans.

#### 3.2.2. Electrophoretic Analysis

The protein profiles of the ME/M ingredient and M powder were analyzed by SDS-PAGE under reducing conditions (Figure 2). Protein identification was based on the standard for bovine caseins (SC pattern) and the previously published literature data for goat milk samples [15,23,32]. As can be seen, known protein bands of goat milk corresponding to caseins and whey proteins can be observed on the control M pattern. On the other hand, in addition to typical and well-known protein bands, three new (non-characteristic) polypeptide bands (marked with *, Figure 2, ME/M pattern) and the band of para-κ-CN can be observed in the M/ME pattern, which was not detected in the control M pattern. In addition, on the same electrophoretic pattern (ME/M), a reduced intensity of the band corresponding to β-CN and the almost complete absence of the k-CN band can be observed. The decreasing intensity of the β-CN and κ-CN bands in the elelectrophoretic pattern of the ME/M ingredient may be due to the proteolytic activity of specific *A. aegerita* enzymes capable of selectively cleaving goat milk caseins. The para-κ-CN detected on the gel originates from κ-CN [32], while the other uncharacteristic bands (marked with *) are probably polypeptides released after the partial proteolysis of β-CN. Previous studies have also shown the proteolytic activity of various mushroom extracts [15,42]. In addition, the aqueous extract of *A. blazei* has been shown to have similar proteolytic activity and the ability to partially or completely cleave β-CN, αs_2_-CN, and γ-CN [15]. In conclusion, the optimized ME/M ingredient exhibits an altered protein profile, with new polypeptides that may contribute significantly to its techno-functional biological and microbiological properties.

#### 3.2.3. ATR-FTIR

The samples (optimized ME/M ingredient and control M powder) were analyzed by ATR-FTIR to (I) confirm the composition of the optimized ingredient; (II) identify changes in the chemical composition as a result of proteolysis or interactions between glucans and proteins; and (III) determine the stability of the obtained ingredient. As can be seen in Figure 3, the ATR-FTIR spectra of the ME/M ingredient and M powder are very similar. In the spectra of both samples, the bands of predominant milk constituents are clearly visible, especially lactose (bands around 1030 cm^−1^) and milk proteins (bands around 1647 cm^−1^ and 1542 cm^−1^) (Figure 3). No new bands or shifts were observed in the overall ME/M spectra, which would indicate proteolysis or glucan–protein interactions (Figure 3). These spectra are similar to the ATR-FTIR spectra previously reported for spray-dried goat milk powders with pollen [24], grape seed extract [22], freeze-dried goji berry extract [43], and *A. blazei* extract [15]. To observe the effect of the added mushroom extract in milk more precisely, the additional statistical processing of the ATR-FTIR spectra and/or deconvolution of some ATR-FTIR regions is required. In this case, the ATR-FTIR spectra screened the prepared ingredient and gave information about predominant compounds in optimized and control samples.

#### 3.2.4. DLS Measurements

The measured particle size values of the control M ranged from 276.7 to 316.7 nm (mean 294.4 ± 17 nm), with an unimodal particle size distribution and polydispersity indices (PI) below 0.3 (Figure 4, see line M). Slight variations in the size of the M particles may be a consequence of the formation of different WP/CN complexes, which are uniformly distributed on the surface of the casein micelles [32,44]. However, the mean particle diameter of the M sample is consistent with data in the literature for thermally treated goat milk [22,45]. On the other hand, the ME/M sample showed a polymodal particle size distribution. In fact, the polydispersity of the sample was so pronounced that no reproducible values could be obtained, as is usual in DLS measurements of highly polydisperse samples. A randomly selected “size distribution” is shown in Figure 4, line ME/M. From the results, it can be concluded that the optimized ME/M ingredient is not homogeneous and contains particles of different sizes. The wide range of the particle diameter of ME/M may be due to intensive proteolysis and the formation of different polypeptides, which are clearly visible in the electrophoretic SDS-PAGE pattern (Figure 2, see ME/M pattern). Soluble para-k-casein and the other polypeptides obtained by proteolysis, as well as modified and partially “stripped” casein micelles, probably contribute to polymodal particle size distribution. In addition, previous studies have shown that the concentration, molecular weight, linkage type, and degree of branching of glucans strongly influence the binding affinity/interactions with milk proteins [46,47]. Glucans from the mushroom extract are likely to interact with soluble polypeptides and WP/CN complexes at the surface of partially “stripped” casein micelles [47] and form complexes of different sizes [46], as shown in the ME/M distribution curves (Figure 4). The absence of the bands that correspond to glucan complexes with caseins, WP/CN complexes, or soluble polypeptides was not observed on the SDS-PAGE gel (see Appendix A), probably due to strongly reducing and dissociating conditions.

### 3.3. SEM of Optimized ME/M Ingredient

Scanning Electron Microscopy (SEM) revealed the morphological features of the optimized ME/M ingredient (Figure 5a–c). As can be seen in the SEM images, the ME/M particles exhibit different and irregular structures that look like flat, rugged, crumpled, and torn flakes or broken porous glass. Figure 5b,c shows particles that are typical of freeze-dried milk powder and look like flat flakes or broken glass [15,48,49]. This morphology is caused by the accumulation of milk proteins over carbohydrates (lactose) on the surface of the protein/carbohydrate [49]. In addition, the microstructure of the protein matrix and the microparticles, which look like small torn flakes with tiny voids, probably indicate proteolytic activity [15,50], as shown by the electrophoretic results. In Figure 5a,b ridge-shaped, rugged, irregular, and aggregated particles are clearly visible (see arrows), indicating the presence of mushroom glucans probably adhering to the surface of the flat-like protein particles.

### 3.4. Techno-Functional Properties of the Optimized ME/M Ingredient

The results of the emulsion activity and emulsion stability of 0.1% ME/M and M solutions are shown in Figure 6a,b. As can be seen, both solutions showed the ability to form and stabilize an emulsion. Both solutions showed good emulsion activity with no significant differences (Figure 6a), primarily due to the ability of the casein micelles and polypeptides to readily coat the surface of small oil droplets [22]. In the case of the ME/M sample, the presence of mushroom glucans also has an influence on emulsion activity. Both solutions also showed good emulsion stability (Figure 6b), which can be attributed to the compact arrangement of the casein micelles forming a uniform interfacial film around the oil droplets. However, the emulsion stability of ME/M decreased significantly over time (from 10 min to 30 min), probably due to the presence of various compounds (glucans, polysaccharides, small molecules) that hinder the formation of a compact film at the oil/water interface and cause the flocculation of the oil droplets. In addition, non-adsorbed glucans and polysaccharides also cause the depletion of flocculation and reduce the stability of the emulsion [51].

The results for foam capacity and foam stability of 0.1% ME/M and the control M solutions are shown in Figure 6c,d. As can be seen, M/ME showed a high capacity for foam formation and stabilization, while the control M solution showed negligible foam capacity and no foam stabilization. It can be concluded that the addition of the mushroom extract and the partial proteolysis of milk proteins have a significant influence on the foaming properties. Mushroom glucans and polysaccharides, together with casein micelles and soluble polypeptides, probably form a compact surface film at the air/water interface. Previous studies have shown that the presence of polysaccharides and conjugates with proteins contributes to foam stability [52]. Milinčić et al. [22] previously reported on the poor foaming properties of thermally treated goat’s milk, which is due to inflexible casein micelles that are unable to reduce the interfacial air/water tension.

### 3.5. Biological Properties of the Optimized ME/M Ingredient

The biological properties of the optimized ME/M ingredient are summarized and illustrated in Figure 7.

#### 3.5.1. Antiproliferative Effect on Human Cancer Cell Lines

The results on the antiproliferative effect of the optimized ME/M ingredient against four cancer cell lines: MCF7 (breast cancer), SiHa (cervical cancer), HepG2 (liver cancer), and Caco-2 (colorectal cancer) are shown in Figure 7a. The antiproliferative effect is measured by the IC_50_ value, which indicates the concentration of ME/M required to inhibit 50% of the cell population, with lower IC_50_ values signifying higher antiproliferative activity. Among the cell lines tested, ME/M showed the highest antiproliferative activity against the Caco-2 colorectal cancer cell line (177.58 µg/mL), followed by moderate activity in the MCF7 breast cancer cell line (198.15 µg/mL). In the SiHa cervical cancer cell line, the ME/M showed a lower antiproliferative activity, with an IC_50_ value of 207 µg/mL, whereby this activity is classified as weak. In the HepG2 liver cancer cell line, the M/ME showed the lowest antiproliferative activity with an IC_50_ value of 296.77 µg/mL. The results obtained show that ME/M had antiproliferative activity towards all the cancer cell lines tested, especially against MCF7 and Caco-2 cells. Previous studies have shown the good anticancer potential of edible mushrooms (including *A. aegerita*) against breast, lung, and colon cancer [53,54,55] and the beneficial effects of goat’s milk and its bioactive peptides against the HCT-116 colorectal carcinoma cell line [56,57]. The good antiproliferative activity of the ME/M ingredient is probably the result of a synergistic effect of the bioactive compounds of *A. aegerita* and the proteins/polypeptides of goat’s milk. In view of these results, the M/ME ingredient could be a promising candidate for further investigation as a potential anti-cancer agent.

#### 3.5.2. Cytotoxicity on Non-Tumor HaCaT Cells

Figure 7b shows the relative growth rate of HaCaT cells exposed to different concentrations of the ME/M ingredient compared to an untreated control group with a relative growth rate of 100%. The results indicate that the M/ME ingredient generally promotes cell growth at higher concentrations, with the effect being particularly pronounced at concentrations of 50 µg/mL and 200 µg/mL. The observed effects of the ME/M ingredient on the proliferation of HaCaT cells may have broader implications for the promotion of epithelial cell growth and repair in various tissues, for example, esophageal cells. Such properties could be advantageous for tissue repair in areas experiencing damage or injury, including those affected by conditions such as chronic irritation or inflammation. Given that the ME/M ingredient is edible, it offers a practical approach for incorporating these bioactive compounds into the diet, potentially supporting natural healing processes from within the body. Oral administration ensures targeted delivery of the active compounds, optimizing their potential for therapeutic effects. However, further research should be conducted to support this thesis.

#### 3.5.3. Wound-Healing Properties

The wound-healing properties of the ME/M ingredient in a HaCaT cell model were determined by measuring the percentage of wound closure. The control group represents the baseline value for wound closure without any treatment with the ingredient. The ME/M ingredient showed a percentage wound closure of 20.24%, which is significantly higher than that of the control (Figure 7c). The good wound-healing quality of the ME/M ingredient is probably due to the mushroom extract [13,58,59], especially the glucans and/or polysaccharides, which have pluripotent characteristics and promote macrophage infiltration and re-epithelization [59,60,61]. This result suggests that ME/M is effective in promoting cell migration and wound closure.

#### 3.5.4. Anti-Inflammatory Properties

The results of the anti-inflammatory activity are shown in Figure 7d. IL-6 levels were measured in picograms per milliliter (pg/mL) and provided insight into the inflammatory response in HaCaT cells triggered by a bacterial infection and the effects of different treatments. The baseline IL-6 level was relatively low (2.96 pg/mL), indicating minimal inflammatory activity in the absence of stimulation. Exposure to *S. aureus* significantly increased IL-6 production in HaCaT cells to 16.64 pg/mL. This substantial increase indicates that *S. aureus* is a potent stimulator of IL-6, reflecting an acute inflammatory response. The addition of the ME/M ingredient (11.31 pg/mL) to inoculated HaCaT cells resulted in reduced IL-6 levels compared to inoculated HaCaT cells with *S. aureus* only, indicating a dose-dependent effect on the inflammatory response. The data suggest that the treatments possess anti-inflammatory properties, effectively reducing IL-6 production in response to the stimulation by *S. aureus*. Previous individual studies have shown the high anti-inflammatory potential of mushrooms [62,63], including *A. aegerita* [13,55,64], as well as goat’s milk [65,66]. However, the results of this study have shown that the combination of the mushroom extract and goat’s milk in the form of the ME/M ingredient may be relevant to therapeutic strategies for the treatment of infections and inflammation associated with *S. aureus*. Further research is needed to understand the mechanisms behind the reduced IL-6 levels in these treatments and their potential clinical applications.

### 3.6. Antimicrobial Activity of Optimized ME/M Lyophilized Ingredient

#### 3.6.1. Antibacterial Activity

The results of antibacterial activity (Table 7) show that among the bacteria tested, *B. cereus*, *S. Typhimurium,* and *E. cloacae* were the most susceptible to the activity of the ME/M ingredient (with MIC values in the range of 0.88–1.75 mg mL^−1^ and MBC in the range of 1.75–3.50 mg mL^−1^). The *S. aureus* strain was the most resilient to the samples tested, with an MBC of over 7.00 mg mL^−1^. Overall, the ME/M ingredient showed better or similar activity compared to the commercial preservatives E211 and E224 used in the food industry. Moreover, the ME/M ingredient showed better antibacterial activity than the *A. aegerita* mushroom extracts without goat’s milk [13], probably due to the presence of bioactive proteins/polypeptides from goat’s milk, which are recognized as promising antimicrobial substances [67,68,69,70]. Regarding the bacterial contaminants identified in the milk (Table 7), the obtained results for the ME/M ingredient were comparable to those obtained for the individual mushroom extracts tested [13]; however, the ME/M ingredient showed slightly better antibacterial potential. Among the tested bacteria, *L. monocytogenes* strains, especially *L. monocytogenes* ATCC 15313 and ATCC 35152 (MIC = 0.44 mg mL^−1^ and MBC = 0.88 mg mL^−1^) were more sensitive to the activity of the ME/M ingredient than *Y. enterocolitica* and tested *Escherichia* spp.

#### 3.6.2. Antifungal Activity

The results of the antifungal activity (Table 8) of the ME/M ingredient were better than its antibacterial activity. A similar trend was observed in our earlier antifungal activity assay when the antifungal activity of mushroom extracts without goat’s milk was tested [13]. The optimized ME/M ingredient showed quite similar activity against the pathogenic fungi tested. However, *P. verrucosum* var. *cyclopium*, *T. viride,* and *A. fumigatus* were the most susceptible (MIC = 0.88 mg mL^−1^, and MFC = 1.75 mg mL^−1^). The values obtained in this study are comparable to those of the positive controls (E221 and E224) (Table 8). In terms of anticandidal potential, the ME/M ingredient showed promising potential with MIC activity in the range of 0.22–0.44 mg mL^−1^ and MFC in the range of 0.44–0.88 mg mL^−1^ (Table 8). The optimized ME/M ingredient showed good antifungal activity, probably due to the synergistic effect of *A. agrocybe* biocompounds, glucans, and goat milk polypeptides, as previously reported in the literature [28,71,72,73]. The results obtained showed a promising effect of ME/M in the control of this opportunistic pathogen. Even if they are several times less effective than ketoconazole, this does not affect their potential to create effective supplements with anticandidal properties.

#### 3.6.3. Antibiofilm Activity

The results of the antibiofilm activity of the ME/M ingredient are shown in Figure 8. The results show that activity was achieved at all three concentrations tested (MIC, one-half MIC, and one-fourth MIC), but not as a function of dose, as the one-fourth MIC value showed a higher potential to disrupt this rigid formation than the higher concentration applied (MIC). The efficiency of sub-inhibitory concentrations in preventing the first initial microbial adhesion, which is a prerequisite for biofilm formation, was previously demonstrated by Cerca et al. [74].

## 4. Conclusions

This study promotes the use of experimental design in the formulation of food ingredients. Therefore, the aim of this study was to develop functional food ingredients combining skimmed goat’s milk and the aqueous mushroom extract of *Agrocybe aegerita* (V. Brig.) Vizzini (ME/M) in an optimal ratio, using CCD as part of RSM. The influence of the two independent variables (pH and mushroom content) on the total phenolic content (TPC), antioxidant activity (ABTS^•+^, DPPH^•^ and FRP assays), and chelating ability (CHE) was investigated. The optimized ME/M ingredient was characterized by ATR-FTIR, electrophoresis, and SEM. This ingredient contains glucans and new polypeptides obtained by the cleavage of β-CN and κ-CN. Due to the different constituents, ME/M showed a polymodal particle size distribution. The 0.1% solutions of ME/M exhibited favorable emulsifying and foaming properties, enabling future application in the formulation of various food products. The newly formulated ME/M ingredient showed good antiproliferative activity against four cancer cell lines, especially Caco-2 colorectal and MCF7 breast cancer cell lines. In addition, ME/M promoted the growth of HaCaT cells, especially at higher concentrations, without being cytotoxic. It was characterized as a promising wound-healing agent in the keratinocyte model. The ME/M ingredient showed the potential to reduce the *S. aureus*-induced inflammation caused in HaCAT cells. The optimized ME/M ingredient showed antibacterial activity, especially against *B. cereus*, *S. Typhimurium*, and *E. cloacae*, as well as antifungal, and anticandidal activity towards all the tested strains. This ingredient also showed antibiofilm one-fourth MIC) and dose-dependent activity. Summarizing the obtained results, the ME/M ingredient has good structural, techno-functional, biological and antimicrobial properties that allow it to be used in a variety of food products, such as dehydrated cream soups, to improve their functionality and shelf-life. In addition, this study can be a promising model for the development of ingredients in the food industry or dietary supplements.

## Figures and Tables

**Figure 1 foods-14-01056-f001:**
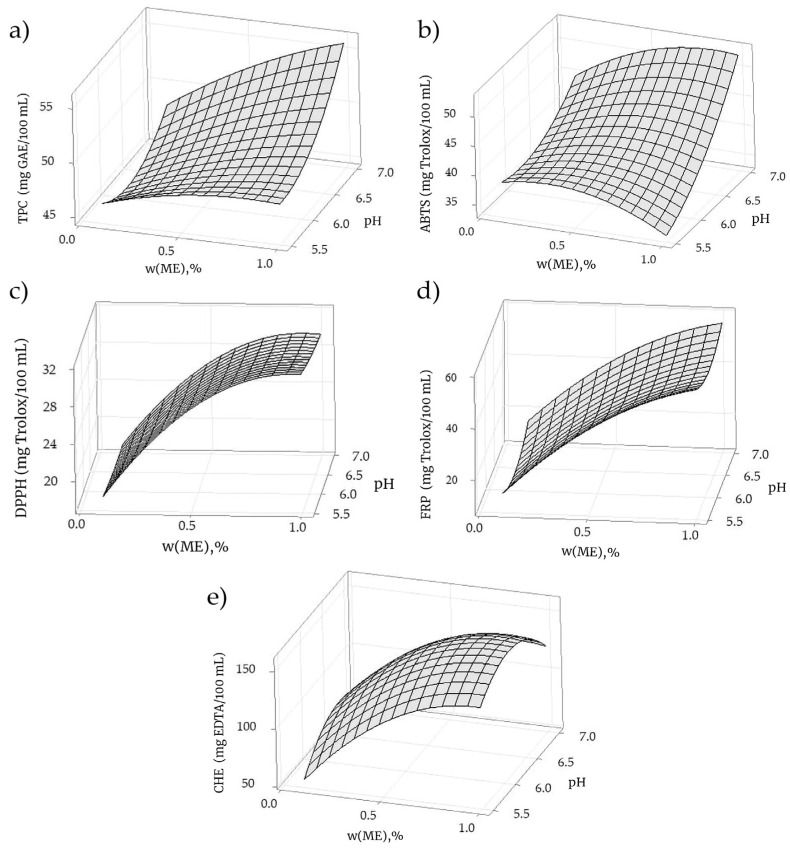
Response surface plot for (**a**) total phenolic content—TPC; (**b**) ABTS^•+^ radical scavenging activity; (**c**) DPPH^•^ radical scavenging activity; (**d**) ferric-reducing power—FRP; and (**e**) ferric-chelating ability—CHE.

**Figure 2 foods-14-01056-f002:**
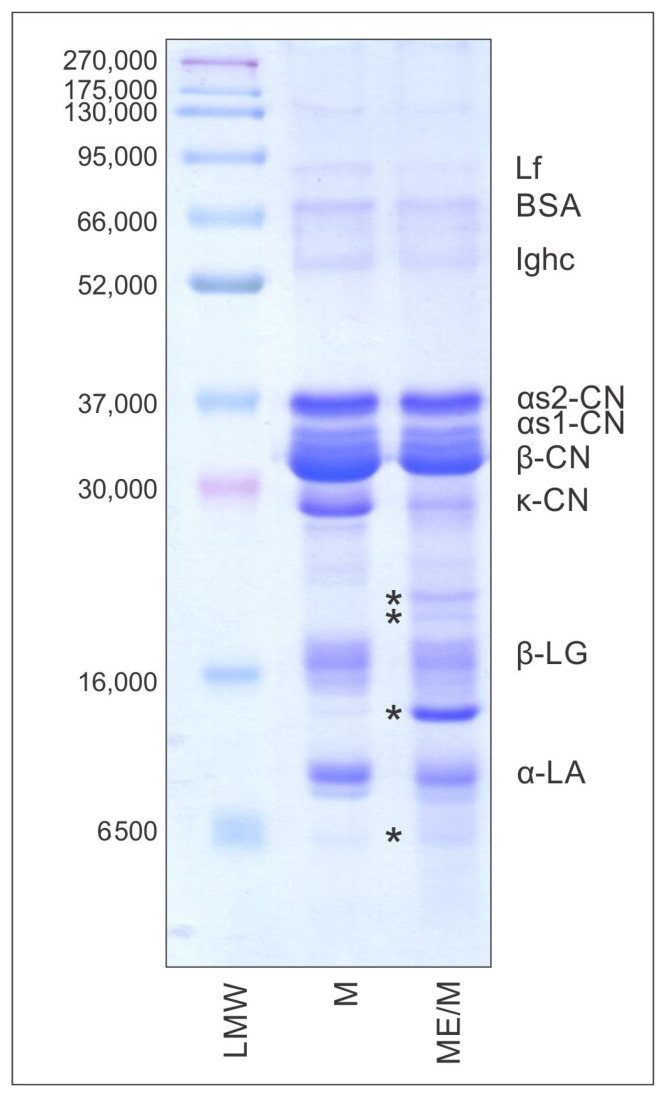
Electrophoretic patterns of the optimized ME/M ingredient and the control M powder analyzed by SDS-PAGE under reducing conditions (SDS-R-PAGE). Molecular weight standard (LMW). Abbreviations: lactoferrin (Lf); bovine serum albumin (BSA); immunoglobulin hard chain (Ighc); αs2-casein (αs2-CN); αs1-casein (αs1-CN); β-casein (β-CN); κ-casein (κ-CN); paraκ-casein (paraκ-CN); β-lactoglobulins (β-LG); α-lactalbumins (α-LA). * Non-characteristic polypeptide bands (products of enzymatic hydrolysis).

**Figure 3 foods-14-01056-f003:**
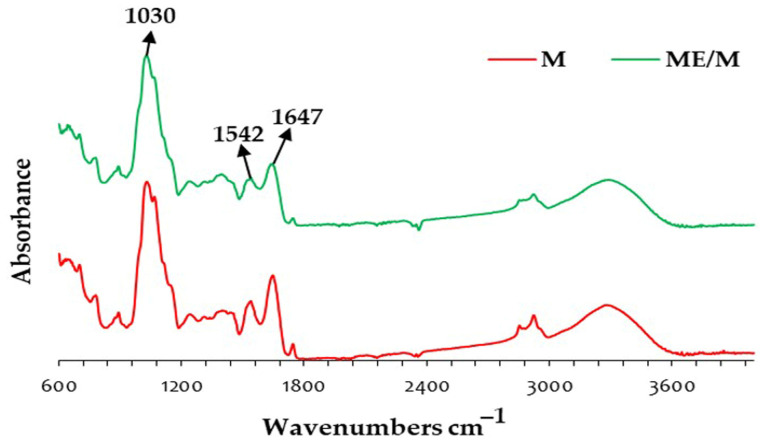
ATR-FTIR spectra of the optimized ME/M ingredient and the control M powder.

**Figure 4 foods-14-01056-f004:**
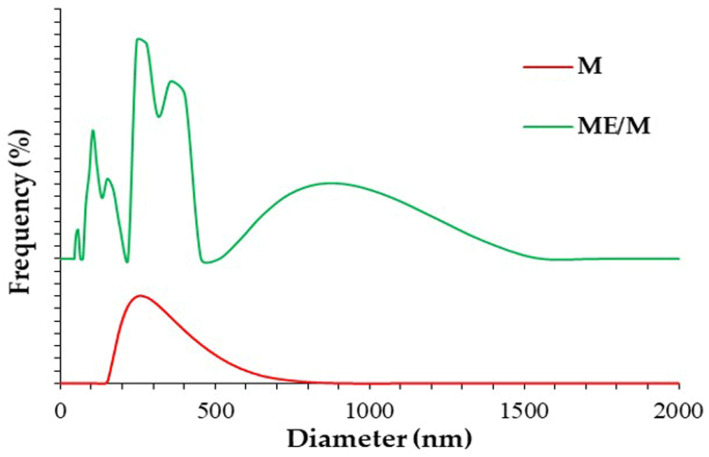
Particle size distribution of the optimized ME/M ingredient and the control M powder.

**Figure 5 foods-14-01056-f005:**
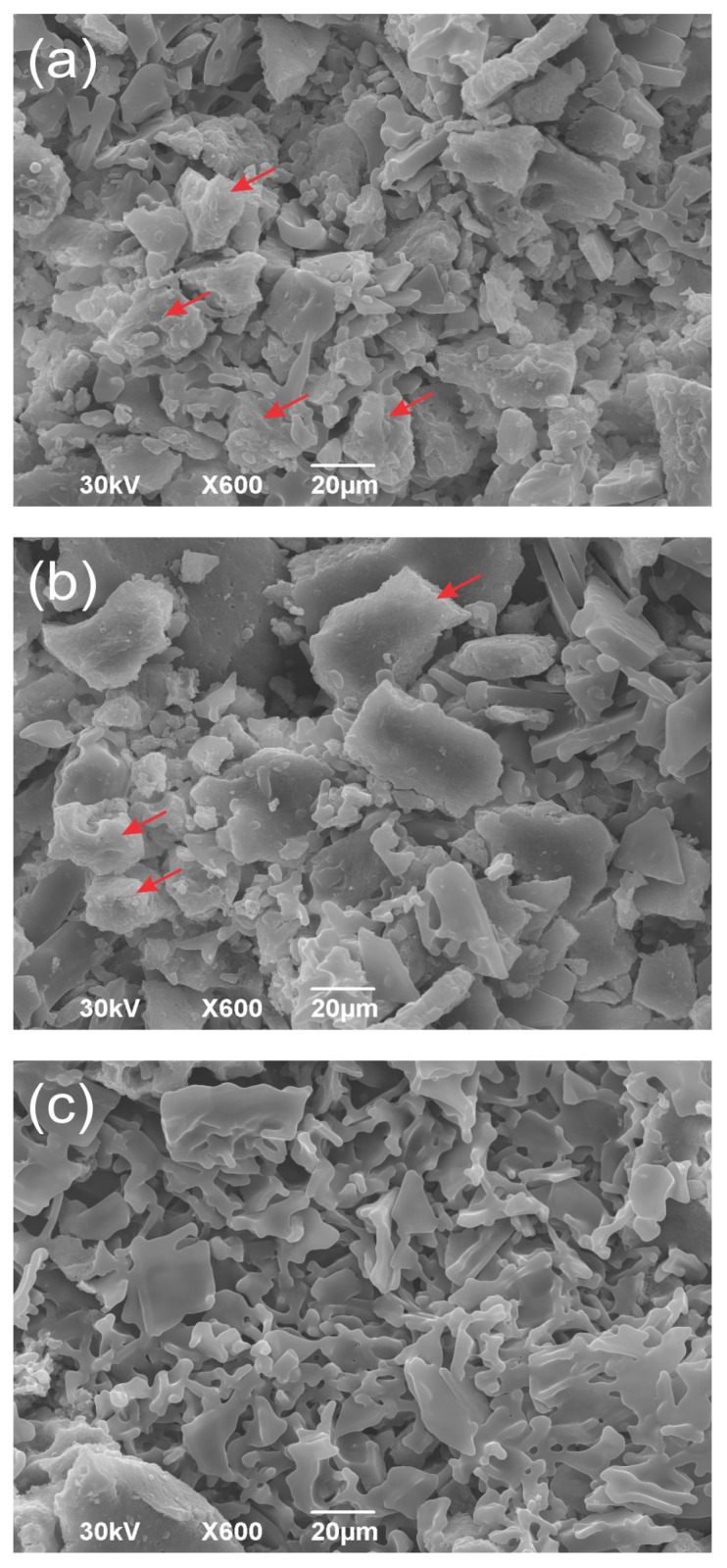
SEM images of the optimized ME/M ingredient (**a**–**c**). Red arrows indicate the presence of mushroom glucans.

**Figure 6 foods-14-01056-f006:**
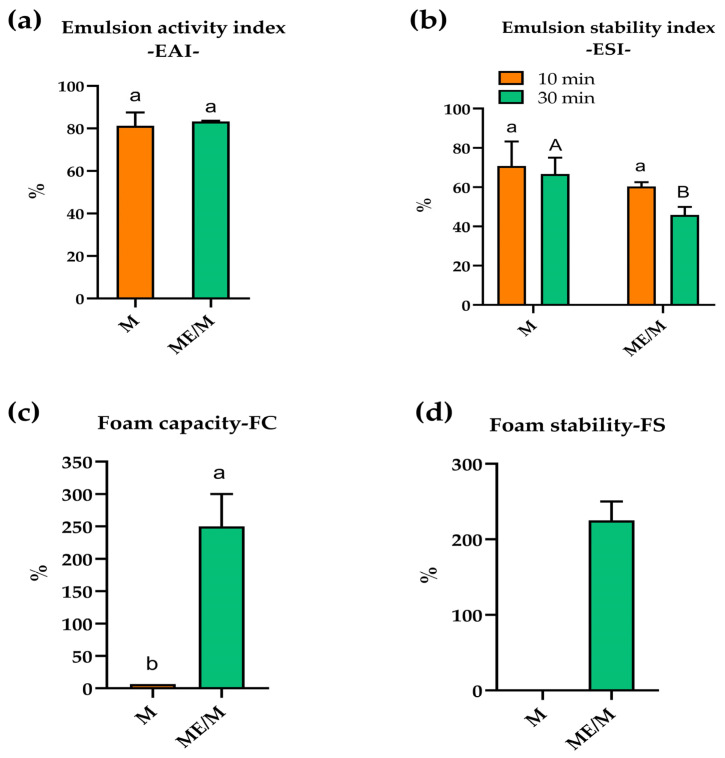
Techno-functional properties of the optimized ME/M ingredient: (**a**) emulsion activity index—EAI; (**b**) emulsion stability index—ESI; (**c**) foam capacity—FC; and (**d**) foam stability—FS. The different lowercase and uppercase letters indicate statistical differences at *p* < 0.05.

**Figure 7 foods-14-01056-f007:**
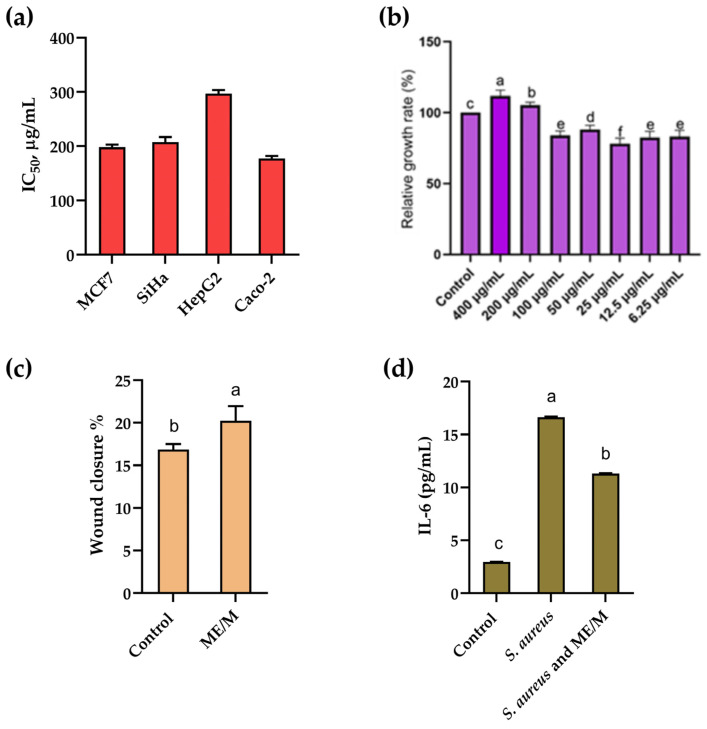
Biological properties of the optimized ME/M ingredient: (**a**) antiproliferative properties; (**b**) cytotoxicity; (**c**) scratch-wound-healing properties; and (**d**) anti-inflammatory properties. The different lowercase letters indicate statistical differences at *p* < 0.05.

**Figure 8 foods-14-01056-f008:**
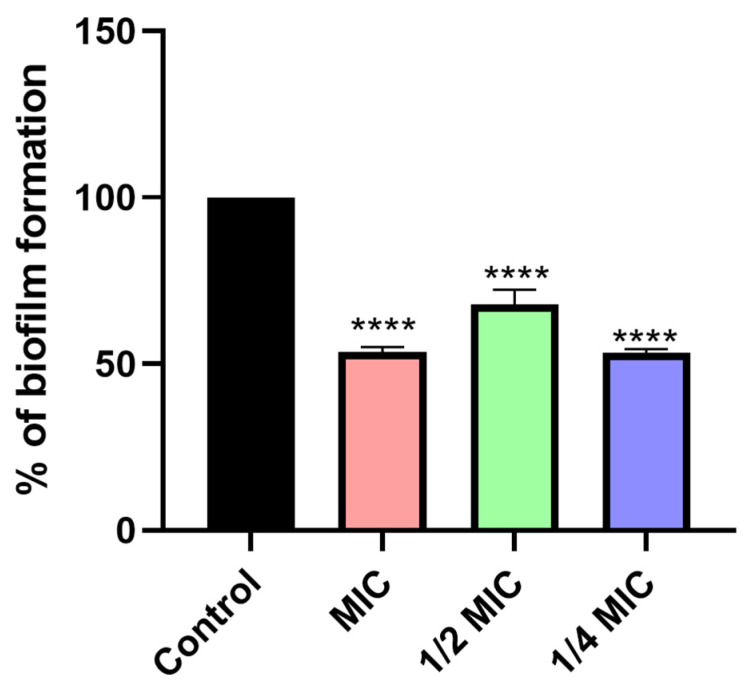
Antibiofilm activity of the optimized ME/M ingredient (% of biofilm formation). The asterisks represent statistical significance **** *p* < 0.0001; the data are presented as mean ± SD.

**Table 1 foods-14-01056-t001:** Experimental factors and levels of factors.

Factor	−1 (Low)	0 (Medium)	+1 (High)
*w* (ME), % (*m*/*m*)	0.1	0.55	1.0
pH	5.5	6.25	7.0

**Table 2 foods-14-01056-t002:** Central Composite Design and real values.

Exp.	*w* (ME), %	pH
1	0.10	7.00
2	0.55	6.25
3	0.55	6.25
4	0.10	5.50
5	0.10	6.25
6	0.55	6.25
7	0.55	6.25
8	0.55	5.50
9	0.55	7.00
10	1.00	6.25
11	1.00	5.50
12	1.00	7.00
13	0.55	6.25

**Table 3 foods-14-01056-t003:** Second-order polynomial equations and regression coefficient of the response variables.

Response	Regression Equation	*R*^2^ (%)
TPC (mg GAE/100 mL)	y = 48.25 + *ε*	56.09
ABTS^•+^ (mg Trolox/100 mL)	y = 42.49 + 6.13∙pH + *ε*	79.19
DPPH^•^ (mg Trolox/100 mL)	y = 27.41 + 6.13∙*w* (ME) − 3.61∙(*w* (ME))^2^ + *ε*	95.61
FRP (mg Trolox/100 mL)	y = 36.94 + 20.86∙*w* (ME) − 6.90∙(*w* (ME))^2^ + *ε*	96.00
CHE (mg EDTA/100 mL)	y = 142 + 34.57∙*w* (ME) − 21.44∙(*w* (ME))^2^ − 25.94∙(pH)^2^ + *ε*	95.50

**Table 4 foods-14-01056-t004:** Data set for overall optimization.

Response	Goal	Lower	Target	Upper
TPC	Target	44	48	57
ABTS^•+^	Maximum	33	51	51
DPPH^•^	Maximum	16	25	25
FRP	Maximum	12	45	45
CHE	Maximum	53	135	135

**Table 5 foods-14-01056-t005:** Predicted and experimental results of analyzed responses.

Response	Individual Desirability	Predicted Value	Obtained Value	Recovery, %
TPC (mg GAE/100 mL)	0.672698	50.95	92.83	182
ABTS^•+^ (mg Trolox/100 mL)	0.759263	46.67	77.79	166
DPPH^•^ (mg Trolox/100 mL)	1.000000	28.90	16.15	56
FRP (mg Trolox/100 mL)	0.999363	44.98	45.34	101
CHE (mg EDTA/100 mL)	1.000000	143.6	211.8	147

**Table 6 foods-14-01056-t006:** Glucan content in optimized ME/M ingredient (% *w*/*w*).

Samples	Total Glucan	α-Glucan	Β-Glucan
	(% *w*/*w*)		
M powder	n.d.	n.d.	n.d.
ME/M ingredient	26.62 ± 0.19	1.94 ± 0.09	24.67 ± 0.10

Abbreviations: n.d.—not detect. M—control thermally treated milk; ME/M—optimized mushroom/milk ingredient.

**Table 7 foods-14-01056-t007:** Antibacterial activity of optimized ME/M ingredient (mg/mL).

Bacteria Strains	ME/M Ingredient		Positive Control			
			E211		E224	
	MIC	MBC	MIC	MBC	MIC	MBC
*Staphylococcus aureus* (ATCC 11842)	3.50	>7.00	4.00	4.00	1.00	1.00
*Bacillus cereus* (clinical isolate)	1.75	3.50	0.50	0.50	2.00	4.00
*Listeria monocytogenes* (NCTC 7973)	1.75	3.50	1.00	2.00	0.50	1.00
*Salmonella Typhimurium* (ATCC 14411)	1.75	3.50	1.00	2.00	1.00	1.00
*Escherichia coli* (ATCC 25922)	1.75	3.50	1.00	2.00	0.50	1.00
*Enterobacter cloacae* (ATCC 45040)	0.88	1.75	2.00	4.00	0.50	0.50
Bacterial contaminants in milk						
*Listeria monocytogenes* (ATCC13932)	0.88	1.75	0.50	1.00	0.50	1.00
*Listeria monocytogenes* (ATCC 15313)	0.44	0.88	0.50	1.00	1.00	2.00
*Listeria monocytogenes* (ATCC 19111)	0.88	1.75	1.00	2.00	1.00	2.00
*Listeria monocytogenes* (ATCC 35152)	0.44	0.88	0.50	1.00	1.00	2.00
*Yersinia enterocolitica* (ATCC 9610)	0.88	1.75	1.00	2.00	0.50	1.00
*Escherichia coli* (ATCC 11775)	1.75	3.50	0.50	1.00	0.50	1.00
*Escherichia coli* O157:H7 (ATCC 700728)	0.88	1.75	1.00	2.00	0.50	1.00
*Escherichia coli* O157:H7 (ATCC 43888)	0.88	1.75	0.50	1.00	1.00	2.00

Abbreviations: MIC—minimal inhibitory concentration; MBC—minimal bactericidal concentration.

**Table 8 foods-14-01056-t008:** Antifungal activity of the optimized ME/M ingredient (mg/mL).

Micromycets	ME/M Ingredient		Positive Control			
			E211		E224	
	MIC	MFC	MIC	MFC	MIC	MFC
*Aspergillus fumigatus* (ATCC 9197)	0.88	1.75	1.00	2.00	1.00	1.00
*Aspergilus versicolor* (ATCC 11740)	1.75	3.50	2.00	4.00	1.00	1.00
*Aspergillus ohraceus* (ATCC 8275)	1.75	3.50	1.00	2.00	1.00	1.00
*Penicillium funiculoseum* (ATCC 48849)	1.75	3.50	1.00	2.00	0.50	0.50
*Penicillium verruscosum* var. *cyclopium* (FI)	0.88	1.75	2.00	4.00	1.00	1.00
*Trichoderma viride* (IAM 5081)	0.88	1.75	1.00	2.00	0.50	0.50
Candida strains	ME/M ingredient		Ketoconazole (×10^−4^)			
	MIC	MFC	MIC		MFC	
*Candida albicans* 475/15	0.22	0.44	4.20		8.40	
*Candida albicans* 14/15	0.22	0.44	1.80		51.20	
*Candida albicans* 17/15	0.44	0.88	1.80		51.20	
*Candida parapsilosis* ATCC22019	0.22	0.44	4.20		8.40	
*Candida tropicalis* ATCC750	0.44	0.88	1.80		8.40	
*Candida krusei* H1/18	0.44	0.88	1.80		4.20	

Abbreviations: MIC—Minimum inhibitory/mycelial growth concentration; MFC—minimum fungicidal concentrations; FI—food isolate.

## Data Availability

The original contributions presented in this study are included in the article/Appendix A, and further inquiries can be directed to the corresponding author.

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
