# Peer review of "Goat’s Skim Milk Enriched with Agrocybe aegerita (V. Brig.) Vizzini Mushroom Extract: Optimization, Physico-Chemical Characterization, and Evaluation of Techno-Functional, Biological and Antimicrobial Properties"

_foods, 2025, doi:10.3390/foods14061056_

Round 1

Reviewer 1 Report

Comments and Suggestions for Authors

  • As authors mentioned in their previous work “Comparative chemical analysis and bioactive properties of aqueous and glucan-rich extracts of three widely appreciated mushrooms: Agaricus bisporus (J.E.Lange) Imbach, Laetiporus sulphureus (Bull.) Murill and Agrocybe aegerita (V. Brig.) Vizzini” (Petrovic et al, 2024); the fruiting bodies of mushrooms were collected in the Zvezdara forest of Belgrade. There is a great variability in the composition of mushrooms growing in different substrates without control of temperature and moist conditions. The mushrooms must be cultured in defined substrates under controlled conditions for better standardization. If the mushrooms can be cultured in vitro they could be provided by certified mushroom farms.

  • I cannot understand the logic of the manuscript, in the first part the authors utilized a central composite design to obtain the best combination of aqueous extract of aegerita (AE) and goat milk but then they used different concentrations of this ingredient (AE/M) (lyophilized?) without reference to the original formulation.

  • The controls of experiments to describe the biological and antimicrobial properties of AE/M ingredient were not properly designed. The authors used PBS to replace AE/M ingredient but should have added other controls: one with goat milk without AE and a condition of only AE,  to check if the effects corresponded to the mushroom extract (AE) or the goat milk (M) or the combination of both.  

  • Based on the results of the poliferative effect of AE/M ingredient on HaCaT cells authors hypothetized that the ingredient would be beneficial in the healing of esophageal injuries. HacaT cells are derived from skin therefore they are very different from esophageal epithelial cells. It is risky to suggest the use of AE/M ingredient for this purpose There is a very long distance between in vitro experiments with a culture cell line and a real effect in vivo.
  •  
  • The italics needed for gender and spices are missing in all the manuscript.

Author Response

The answers to reviewer comments is in attachment.

Reviewer 2 Report

Comments and Suggestions for Authors

The manuscript “Goat skim milk enriched with Agrocybe aegerita (V. Brig.) Vizzini mushroom extract: Optimization, physico-chemical characterization, and evaluation of techno-functional and functional properties” could be interesting but it needs to be improved, especially the results need to be strengthened with more discussions and comparison with literature and by adding the practical impact of the research. 

Following some suggestions.

What is the difference between techno-functional and functional properties?

Line 51: check the sentence

Line 58: check gycopeptides and biocompounds

Line 78: check the sentence

Line 82:  check the sentence

The authors need to introduce more information on Agrocybe aegerita in the text. Why did they choose this mushroom?

Did the authors take into account that also goat milk has distinctive sensory features?

The authors indented to use this food ingredient for what kind of food product?

Line 103: ingredient

Line 148: check Folin-Ciocalteu

Par. 3.1 The aim of the central composite design has been already described.

How did the authors explain the contrasting results related to TPC and DPPH?

Par. 3.2 The authors should include in the Figure the electrophoretic pattern of goat milk and the aqueous extract of mushrooms, to have a reference. How did the authors explain the decrease of k-Cn band in the ME/M mixture and probably the formation of para-K-casein?

Line 497-499. This explanation is not clear. Can the polypeptides affect the particles size? Moreover, any aggregates had to be visible also at the top of the SDS-PAGE. Could it be considered that the solubilization procedures should be improved? Also, because, it seems that coagulation event occurred.

Comments on the Quality of English Language

English grammar should be improved.

Author Response

The answers to reviewer comments are in the attachment.

Round 2

Reviewer 1 Report

Comments and Suggestions for Authors

Manuscript number Foods 3502354v2

Here are my suggestions:

Major:

  • The manuscript does not have a clear motivation. What is the future use of the new ingredient M/ME? I think the title does not correspond with the work. The authors do not design a supplemented goat milk as the title said. They create a new ingredient for an unknown purpose.

  • In the physicochemical characterization of optimized ME/M ingredient, the authors used as a control goat milk. However, in all the biological experiments, the control was PBS. The appropriate control for these experiments should contain only milk, aqueous mushroom extract and the mix of both ingredients in the correct proportions. It is not adequate to compare the effect obtained previously with the aqueous extract of aegerita with the results obtained for the ME/M ingredient in this work. The content of mushroom extract is not equivalent between both formulations.

Minor:

  • Correct the spelling of characterization, lyophilized.

Author Response

Reviewer 1.

We understand your concerns. All comments have been addressed below, and the corresponding corrections have been written in green in the manuscript.

Major:

  • The manuscript does not have a clear motivation. What is the future use of the new ingredient M/ME? I think the title does not correspond with the work. The authors do not design a supplemented goat milk as the title said. They create a new ingredient for an unknown purpose.

This ingredient was formulated to enrich food products in which milk proteins are commonly used as ingredients, such as dehydrated cream soups, to enhance their functionality. This is highlighted in the Conclusion section; see line 748 in the revised manuscript. This type of soup has already been formulated and characterized, and the results are planned to be published in a separate manuscript.

The ingredient contains 10% skimmed goat milk and 0.7% mushroom extract, which is commonly considered an enrichment. The addition of varying percentages of bioactive compounds or extracts to milk or dairy products—often in amounts much higher than in this study—is a well-documented enrichment approach in the literature (Adinepour et al., 2022; Kandylis et al., 2021; Kostić et al., 2021; Milinčić et al., 2021; Milinčić et al., 2024; Milinčić et al., 2025; Minić et al., 2023; Sangsopha et al., 2019; Thakur & Nanda, 2019; Wazzan, 2024). For this reason, the newly formulated ingredient in this study was named “skimmed goat milk enriched with mushroom extract”.

In the physicochemical characterization of optimized ME/M ingredient, the authors used as a control goat milk. However, in all the biological experiments, the control was PBS. The appropriate control for these experiments should contain only milk, aqueous mushroom extract and the mix of both ingredients in the correct proportions. It is not adequate to compare the effect obtained previously with the aqueous extract of aegerita with the results obtained for the ME/M ingredient in this work. The content of mushroom extract is not equivalent between both formulations.

We understand your concerns. The aim of this study was to develop an optimized functional ingredient based on goat milk proteins and mushroom extract using Central Composite Design and Response Surface Methodology to achieve maximum antioxidant properties. After optimization, the ingredient was characterized for its techno-functional, biological, and microbiological properties. In experiments where experimental design is used to optimize parameters for developing products with enhanced targeted functionality, it is common practice to characterize the other functional properties of the optimized product without directly comparing them to those of its individual constituents (Pant et al., 2022; Pranowo et al., 2020; Thakur & Nanda, 2019; Thakur et al., 2021). To better clarify the study’s objective, additional text has been added to the introduction section. See lines 90 and 101–103 in the revised manuscript.

Minor:

  • Correct the spelling of characterization, lyophilized.

Thank you very much for the suggestion. The correction has been made accordingly.

References:

Adinepour, F., Pouramin, S., Rashidinejad, A., & Jafari, S. M. (2022). Fortification/enrichment of milk and dairy products by encapsulated bioactive ingredients. Food Research International, 157, 111212. https://doi.org/https://doi.org/10.1016/j.foodres.2022.111212

Kandylis, P., Dimitrellou, D., & Moschakis, T. (2021). Recent applications of grapes and their derivatives in dairy products. Trends in Food Science & Technology, 114, 696-711. https://doi.org/https://doi.org/10.1016/j.tifs.2021.05.029

Kostić, A. Ž., Milinčić, D. D., Stanisavljević, N. S., Gašić, U. M., Lević, S., Kojić, M. O., Lj. Tešić, Ž., Nedović, V., Barać, M. B., & Pešić, M. B. (2021). Polyphenol bioaccessibility and antioxidant properties of in vitro digested spray-dried thermally-treated skimmed goat milk enriched with pollen [Article]. Food Chemistry, 351, Article 129310. https://doi.org/10.1016/j.foodchem.2021.129310

Milinčić, D. D., Kostić, A. Ž., Gašić, U. M., Lević, S., Stanojević, S. P., Barać, M. B., Tešić, Ž. L., Nedović, V., & Pešić, M. B. (2021). Skimmed goat’s milk powder enriched with grape pomace seed extract: Phenolics and protein characterization and antioxidant properties [Article]. Biomolecules, 11(7), Article 965. https://doi.org/10.3390/biom11070965

Milinčić, D. D., Kostić, A. Ž., Kolašinac, S., Rac, V., Banjac, N., Lađarević, J., Lević, S., Pavlović, V. B., Stanojević, S. P., Nedović, V. A., & Pešić, M. B. (2024). Goat milk powders enriched with grape pomace seed extract: Physical and techno-functional properties [Article]. Food Hydrocolloids, 146, Article 109293. https://doi.org/10.1016/j.foodhyd.2023.109293

Milinčić, D. D., Kostić, A. Ž., Lević, S., Gašić, U. M., Božić, D. D., Suručić, R., Ilić, T. D., Nedović, V. A., Vidović, B. B., & Pešić, M. B. (2025). Goat’s Milk Powder Enriched with Red (Lycium barbarum L.) and Black (Lycium ruthenicum Murray) Goji Berry Extracts: Chemical Characterization, Antioxidant Properties, and Prebiotic Activity. Foods, 14(1), 62. https://www.mdpi.com/2304-8158/14/1/62

Minić, D. A. P., Milinčić, D. D., Kolašinac, S., Rac, V., Petrović, J., Soković, M., Banjac, N., Lađarević, J., Vidović, B. B., Kostić, A. Ž., Pavlović, V. B., & Pešić, M. B. (2023). Goat milk proteins enriched with Agaricus blazei Murrill ss. Heinem extracts: Electrophoretic, FTIR, DLS and microstructure characterization [Article]. Food Chemistry, 402, Article 134299. https://doi.org/10.1016/j.foodchem.2022.134299

Pant, K., Thakur, M., Chopra, H. K., & Nanda, V. (2022). Encapsulated bee propolis powder: Drying process optimization and physicochemical characterization. LWT, 155, 112956. https://doi.org/https://doi.org/10.1016/j.lwt.2021.112956

Pranowo, D., Perdani, C. G., Prihardhini, T. A., Wijana, S., Qmr, A. S. F., & Arisandi, D. M. (2020). Optimization of Microencapsulation Process of Green Coffee Extract With Spray Drying Method as a Dietary Supplement. Pelita Perkebunan (a Coffee and Cocoa Research Journal).

Sangsopha, J., Moongngarm, A., Johns, N. P., & Grigg, N. P. (2019). Optimization of pasteurized milk with soymilk powder and mulberry leaf tea based on melatonin, bioactive compounds and antioxidant activity using response surface methodology. Heliyon, 5(11). https://doi.org/10.1016/j.heliyon.2019.e02939

Thakur, M., & Nanda, V. (2019). Process optimization of polyphenol-rich milk powder using bee pollen based on physicochemical and functional properties. Journal of Food Process Engineering, 42(6), e13148. https://doi.org/https://doi.org/10.1111/jfpe.13148

Thakur, M., Pant, K., Naik, R. R., & Nanda, V. (2021). Optimization of spray drying operating conditions for production of functional milk powder encapsulating bee pollen. Drying Technology, 39(6), 777-790. https://doi.org/10.1080/07373937.2020.1720225

Wazzan, H. (2024). Fortification of Dairy Products using Plant-derived Bioactive Compounds. Current research in nutrition and food science, 12(2), 561-571. https://doi.org/dx.doi.org/10.12944/CRNFSJ.12.2.6

Reviewer 2 Report

Comments and Suggestions for Authors

The authors improved the manuscript according to the suggestions and comments

Author Response

Thank you very much for your feedback and for helping to improve the quality of the manuscript.